# Hollow Silica Nano and Micro Spheres with Polystyrene Templating: A Mini-Review

**DOI:** 10.3390/ma15238578

**Published:** 2022-12-01

**Authors:** Siddharth Gurung, Francesco Gucci, Gareth Cairns, Iva Chianella, Glenn J. T. Leighton

**Affiliations:** 1Surface Engineering and Precision Centre, Department of Manufacturing and Materials, Cranfield University, Bedfordshire MK43 0AL, UK; 2Atomic Weapons Establishment, Reading, Berkshire RG7 4PR, UK

**Keywords:** hollow silica sphere, silica, hard template, sol–gel, shell thickness, inner diameter

## Abstract

Synthesis of monodisperse hollow silica nanospheres, especially using a hard template route, has been shown to be successful, but a high yield is needed for this strategy to be used on an industrial scale. On the other hand, there is a research gap in the synthesis of hollow silica microspheres due to the popularity and easiness of the synthesis of silica nanospheres despite the larger spheres being beneficial in some fields. In this review, current trends in producing hollow silica nanospheres using hard templates, especially polystyrene, are briefly presented. Soft templates have also been used to make highly polydisperse hollow silica spheres, and complex designs have improved polydispersity. The effect of the main parameters on the coating is presented here to provide a basic understanding of the interactions between the silica and template surface in the absence or presence of surfactants. Surface charge, surface modification, parameters in the sol–gel method and interaction between the silica and templates need to be further improved to have a uniform coating and better control over the size, dispersity, wall thickness and porosity. As larger organic templates will have lower surface energy, the efficiency of the micro sphere synthesis needs to be improved. Control over the physical structure of hollow silica spheres will open up many opportunities for them to be extensively used in fields ranging from waste removal to energy storage.

## 1. Introduction

Silica (silicon dioxide) is a naturally occurring material found in rocks and sands in amorphous and crystalline forms. Various engineered forms such as gels have been used in various textile and food industries [1,2,3,4]. A smaller particle in the nanometre and micrometre range has different properties from large bulk materials due to a high surface area to volume ratio. Advances in nanomaterials have allowed for the synthesis of silica spheres that can produce coatings or make composite materials [5,6,7]. The first synthesis method to create silica nanospheres was the Stöber method [8], based on the hydrolysis of alkoxide precursors, which condenses to form a silica network. There has always been interest in hollow silica spheres due to their low weight and highly desirable properties such as high mechanical strength, low electric, and thermal insulation [9,10], but the majority of research has focused on hollow silica nanospheres compared to the micron size. This lack of research in micron size spheres could be due to the difficulty in synthesis and the highly lucrative markets of nanoparticles as fillers in polymers, coatings, and insulation [11]. However, there are several applications for micron-sized hollow spheres, which are described later. This review focuses on the leading strategies for synthesising hollow silica spheres using hard templates (especially polystyrene (PS)) to provide an overview of the most recent discoveries and evidence of the most promising approaches.

### 1.1. Hollow Silica Sphere

As the name suggests, a hollow silica sphere is a particle that has a hollow interior with a solid shell that may or may not be porous. Hollow spheres have different properties than dense bulk particles due to their empty core. The thickness of the shell and its porosity will hugely affect properties such as density, surface area, thermal conductivity, loading capacity, etc. [12,13]. Some applications of hollow silica spheres can be seen in works involving insulation and drug delivery [10,14]. A high degree of control of shell thickness and the hollow interior is needed to achieve a specific behaviour. High polydispersity and low efficiency at a larger scale make manufacturing hollow silica spheres difficult in the micron range. Gravity becomes dominant when particle size increases from nm to µm, while has a negligible effect when particles are tiny (colloids). The low surface energy of the templates discussed below has a considerable impact on the micron range.

### 1.2. Comparison of Synthesis Strategy

There are various strategies to synthesize hollow spheres. J. Sharma et al. [15] reviewed methods currently used to synthesize hollow silica particles (HSPs). Their comparison of different routes to obtaining hollow silica spheres is shown in Table 1 with their advantages and disadvantages. This review explored hard template strategies using sol–gel chemistry to produce hollow silica spheres with an organic template (polymer) as it allows for an easier and cheaper route to synthesize PS spheres. The hard template strategy has its own issue with low yield, but as discussed later, it can be improved. Other related methods are briefly explained where required.

As shown in Table 1, inorganic, organic, and silica templating methods allow for reasonable control in the size and dispersity. Similarly, the emulsion method is often used to synthesize hollow nanospheres, where a coating is obtained at the interface between two phases: a dispersed phase (droplets) and a continuous phase (surrounding liquid), which are created when two immiscible liquids are mixed. In general, a template strategy (Figure 1) utilises solid particles or an emulsion as a template that is coated by silica. A hollow sphere is left behind when the template is removed, either by chemical etching or calcination, and the shell is thick enough to withhold the structure without collapsing. In general, the templating method can be divided into two groups. A schematic diagram of the different synthesis routes is shown in Figure 2 and Figure 3, respectively.

In soft templating, a surfactant and an emulsion of water and oil provide the template, which is then coated with the sol–gel method. Soft template methods are very sensitive due to the instability of the emulsion, so they need strict control of the reaction parameters as products have been reported to be polydisperse and aggregated [8,17,18,19]. Surfactants will be explained in Section 3.2. Instead, hard templating methods use an organic or inorganic solid particle as a seed, allowing for tighter control of the inner diameter. Shell thickness is controlled by the kinetics of the solution reaction and the template surface chemistry [20,21,22,23]. Other methods such as spray drying and the microfluidic method are generally avoided due to being too complex, slow, or leading to a highly polydisperse product. In the last case, an analysis of every single product is needed, which adds to the labour costs. If the yield of the hard templating route can be improved, the cost of hollow silica would drastically decrease. 

**Figure 2 materials-15-08578-f002:**
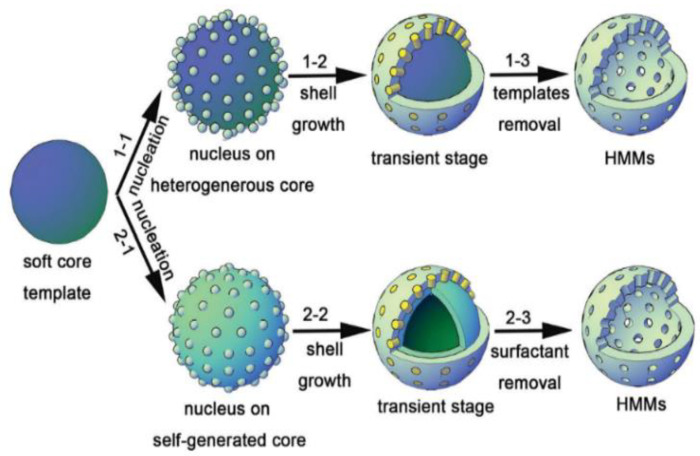
Illustration of the soft template route via (1-1) heterogenous soft-templating route and (2-2) self-generated soft-templating route to produce hollow-structured mesoporous materials (HMMs) [24].

**Figure 3 materials-15-08578-f003:**
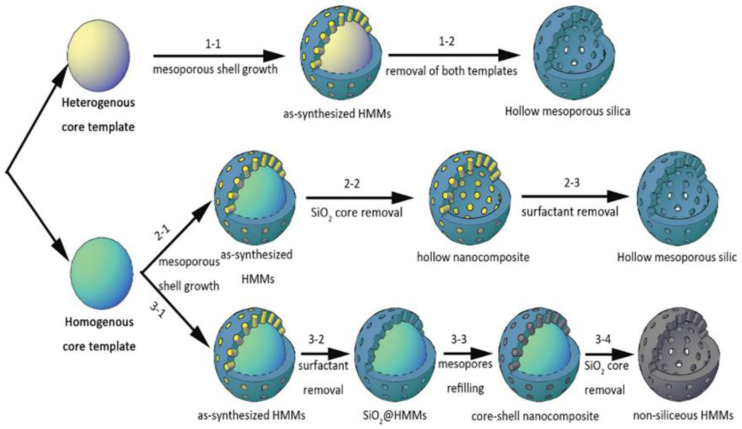
Illustration of the route of hard templating: (1-1) Heterogenous core, (2-1,3-1) homogenous core with the different procedures carried out to produce either hollow mesoporous silica or non-siliceous HMMs [24].

## 2. Sol–Gel

Before reviewing the hard templating route, it is necessary to discuss sol–gel chemistry [25] as it is fundamental to understanding the process. In sol–gel, there are two key components:Sol: Stable suspension of colloidal solid particles in a liquid. The repulsive force allows the small particles to overcome the attractive force and not aggregate. Gravity forces are negligible due to the small size of particles.Gel: A porous three-dimensionally interconnected solid network in a liquid. Based on the solvent nature and drying method, a gel can be converted to a hydrogel, aerogel, alcogel, or xerogel.

The sol–gel method is a liquid state preparation where a sol evolves into a polymer-like network (gel) that consists of a solid and liquid phase. The main advantages of sol–gel are its low operation cost due to the low temperature requirement (*up to* 320 °C) [26,27,28,29] for forming metal oxide bonds, which usually require a high temperature (1400–3600 °C) [30,31,32,33] when traditional methods are used Materials with very high purity can be achieved via sol–gel, which is very useful. One disadvantage is the reaction time as it could last longer than weeks, depending on the reaction conditions. Combined with the proper drying conditions, it is possible to achieve different products such as films, bulk ceramic particles, or aerogels, as shown in Figure 4.

### Sol–Gel Chemistry 

In 1968, Werner Stöber and Arthur Fink used a colloidal suspension to obtain monodisperse and homogenous spherical silica nanoparticles using ammonium hydroxide (NH_4_OH), silicon alkoxide, water, and alcohol [8]. This process, known as the Stöber method, has been modified to produce highly monodispersed or aggregated silica particles [35,36,37]. Depending on the reaction kinetics, a catalyst may be needed to increase the chemical reaction. The reaction of silicon alkoxides will be used as a common reference in this review. Alcohol is required as a cosolvent for silicon alkoxides as they are immiscible in water. Depending on the volume of water and alcohol, the solution is often referred to as aqueous (*high volume of water*) or non-aqueous (*high volume of alcohol*). Usually, the solution is turbid after the reaction due to colloids that scatter light. Surfactants such as cetyltrimethylammonium bromide (CTAB), as discussed in Section 3.2, can improve the surface and solution reactions. NH_4_OH may be referred to as ammonia or NH_3_ in some sections.

The two stages and the chemical reactions involved in the synthesis of silica particles are reported below.


**Hydrolysis stage**
≡Si-OR + H_2_O **⇌** ≡Si-OH + ROH (1)


Forward: hydrolysis, reverse: esterification


**Condensation stage**
≡Si-OR + HO-Si ≡**⇌** ≡Si-O-Si ≡ +ROH(2)


Forward reaction: alcohol condensation, reverse reaction: alcoholysis
≡Si-OH + HO-Si ≡ **⇌** ≡Si-O-Si ≡ + H_2_O(3)

Forward reaction: water condensation, backward reaction: hydrolysis

where R = alkyl group

From the previous equations, hydrolysis of the alkoxide precursor produces a monomer (Si–OH) and related alcohol (ROH) (Equation (1)). Depending on the reaction pathway, these hydrolysed monomers can condense with another monomer or already condensed particles to form a siloxane bond (Si–O–Si) and produce water or alcohol as a secondary product. (Equations (2) and (3)). Precursors could be partially or entirely hydrolysed (Figure 5), determining how well they will condense. Tetraethyl orthosilicate (TEOS) is the most widely used silicon alkoxide, but sodium silicate is also routinely used. Many factors need to be considered when choosing the precursor, which affects its reaction kinetics. Steric hindrance and the polarity of the substituents have been shown to affect the thermodynamics and reaction kinetics. The hydrolysis rate was observed to decrease from tetramethyl orthosilicate (TMOS, TEOS) to tetra propyl orthosilicate (TPOS), while the condensation rate increased from TMOS, TEOS to TPOS. The same effect was seen in different alcohols, with methanol having the highest hydrolysis rate while propan-2-ol had the highest condensation rate [38].

La Mer’s theory describes the nucleation and growth of silica particles [39]. This involves an induction period, supersaturation, nuclei formation, and the growth of particles. The induction period is described as a time when no particles are seen, and the hydrolysis reaction is dominant. As the hydrolysed monomer concentration increases rapidly and starts to aggregate, the reaction will reach a supersaturation concentration. This means that silica aggregates cannot stay in solution anymore as they exceed the solvent capability to dissolve them. The precipitation of silica aggregates from the solvent produces nuclei that grow by condensation of the dissolved silica aggregates. It has been shown that the pH contributes the most to species reactivity compared to other variables such as water, temperature, co-solvent, etc. [40,41]. Y. Han et al. [42] showed that the concentration of ammonia (catalyst) affects the degree of hydrolysis (Figure 5). The mass spectroscopy results showed that the higher the ammonia concentration, the higher the concentration of fully hydrolysed TEOS (Figure 5 (Q (040)). The presence of different hydrolysed TEOS explains the variation of reaction kinetics at different catalyst concentrations. The availability of active sites and the level of steric hindrance in monomers will affect the direction and rate of condensation, significantly affecting the final structure. The interactions between reactants are complex to understand, making it hard to analyse the correlation between them and other parameters. However, it has been shown that the growth of silica particles is proportional to the concentration of NH_3_, H_2_O, and TEOS, and inversely proportional to temperature [41,43,44,45,46,47,48,49,50,51]. On the other hand, it must be considered that there is not a linear rise but a fluctuation as the concentrations or physical parameters are changed. This is due to the interactive effects between parameters, as shown by the statistical approach of R.S. Fernandes et al. [52] and Chiang et al. [40]. This explains the contradictory results from different research projects as the reaction parameters differ.

It is essential to control the reaction kinetics and nucleation as nuclei are generated beyond the induction period. If nuclei formation is high, smaller nuclei will grow and compete with larger particles for smaller particles. This is known as secondary nucleation and increases polydispersity [53,54,55]. Monodispersed spheres are seen at high ammonia concentrations and are proposed due to the suppression of secondary nucleation by highly effective hydrolysis [42]. Fully hydrolysed TEOS (Figure 5 (Q (040)) is very reactive and nuclei formed by this type of TEOS are likely to condense on large particles rather than aggregating. The ineffective hydrolysis of precursors is proposed to be the reason for polydispersity in low ammonia concentration. A high homogenous reaction (i.e., the reaction between free silica particles) produces more nuclei (secondary nucleation), which could grow larger by consuming smaller silica/hydrolysed monomers, thus reducing the coating efficiency. Therefore, a heterogeneous reaction between silica particles/hydrolysed monomers with the template surface must be dominant and homogenous reactions must be suppressed or kept low to achieve a high coating.

**Figure 5 materials-15-08578-f005:**
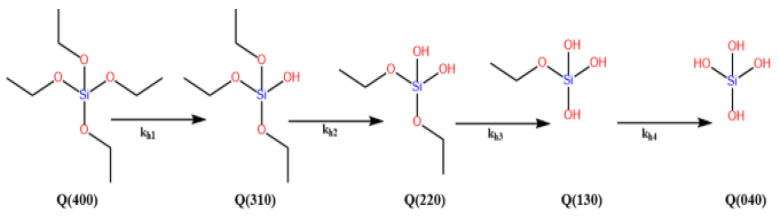
Different degrees of hydrolysed TEOS [56].

## 3. Other Chemicals

As above-mentioned, chemicals used during template synthesis, surface modification, and coating affect the features of the final product. This section briefly describes the most important and commonly used chemicals and their role in synthesising the polymeric template and its modification (i.e., coating) to obtain silica particles.

### 3.1. Initiator and Stabilisers for PS Template

An initiator is usually needed to start the polymerisation process to synthesise the organic template. It provides an intermediatory compound when it reacts with a monomer, thus allowing it to react with other monomers. Such monomers are then linked to the initial structure like a chain of rings. The reaction will continue until the chain is terminated (either by reacting with another chain or another initiator) and the polymer is formed. Different initiators used during synthesis can influence the surface charge [57,58]. Initiators that are routinely used during the synthesis of PS are AIBN (2,2′-azobis(isobutyronitrile)), AIBA (α,α′-azodiisobutyramidine dihydrochloride), and KPS (potassium persulfate). On the other hand, stabilisers are also needed to control the reaction as they promote termination and influence the reaction by steric hindrance, which is vital to control the size and molecular weight of the polymer template produced.

### 3.2. Surfactant

Surfactants are amphiphilic molecules consisting of a head and a tail, which can be hydrophilic and hydrophobic and have been widely used to improve the coating of PS. Their use has been reported to enhance the interaction between templates and silica, thus improving the coating rate [59,60,61]. Depending on their charge, they can be classified as non-ionic, anionic, cationic, and amphoteric (Figure 6). Various groups have suggested that they act as a linker between the surface of the template and the silica particles [62,63,64] and can also act as a nucleation site for silica particles to aggregate in solution.

The surfactant’s aggregated shape and activity are determined by the surfactant’s concentration and solution temperature (Figure 7). As the concentration increases, they will linearly reduce the system’s surface tension until it reaches a critical value, called the critical micelle concentration (CMC), where a surface is saturated with the surfactant. The temperature will affect the concentration at which saturation occurs, as shown in Figure 7. At CMC, they will start to aggregate into different structures such as spherical structures (micelle) and orient toward the main solvent (i.e., all the hydrophilic head toward the water if it is the main solvent). It is unclear how the surfactants can attract the silica particles to the PS surface; the theories range from the deposition of a micelle of silica and surfactant to the templating action of the skeletal backbone of the adsorbed surfactant [53,58]. It has been shown that different concentrations of surfactant produce either higher secondary particles and thinner shells, thicker coating, or no coating at all. Moreover, it has been shown to influence pore distribution, but not pore size [65].

## 4. Research Gap

Table 2 contains information about the chemicals, shell thickness, size of spheres, and pores from various works synthesising hollow silica spheres using PS, which will be described later in the next section. As can be seen, most of the research has focused on particles with diameters in the nanometre range, with very few works on the micrometre. A difficulty arises mainly from the poor surface chemistry of PS as it has low surface energy. The surface area will also increase when the size increases, making the coating more difficult. Therefore, this issue requires some strategy to be overcome such as combining PS with a more reactive polymer or modifying its surface. Surfactants were also found to help in the coating. New methods of improving surface reactions with surface modification and a better understanding of sol–gel chemistry are reviewed to guide future works. It must be noted that soft templates have also been used to successfully synthesise hollow silica microspheres but with high polydispersity. H. Yoon et al. [19] successfully synthesised hollow silica spheres using soybean oil to create a water/oil emulsion. An average diameter of 35.5 micron with a 50.5% coefficient of variation (CV) was achieved, while the shell thickness was reported to increase from 453.2 nm to 549.5 nm, with TEOS rising from 0.25 to 1 g. When combined with microfluidic technology, highly monodispersed sizes have been reported. Li et al. [67] developed a microfluidic device to prepare monodispersed hollow silica microspheres in an emulsion. The emulsion size could be easily controlled by adjusting the flow rate of oil and the aqueous phase. The emulsion was then coated through interfacial polymerisation. The concentration of TEOS and CTAB was responsible for the formation of the morphology and the presence or absence of the hollow core. The diameter of hollow silica spheres was reported to be between 91 and 167 micrometres, with a shell thickness of around 500 nm. W. Jeong et al. [68] successfully controlled the double emulsion’s size by controlling the water ratio to the oil phase and created a double emulsion of 60 microns with a CV below 7%. These emulsions produced a sub-micron hollow silica sphere that was successfully synthesised by altering the parameters of the surfactant, tetrahydrofuran, ethanol, silica sols, and the ratio of oil and water phase. Microfluidic methods use capillary, which can be made of polymer or glass, which have associated issues such as chemical resistance, visibility, cost, and complexity in design [69,70,71,72]. A. Niculescu et al. extensively reviewed the fabrication methods and application of microfluidics for the synthesis of silica particles [73]. Some of the reported disadvantages in the fabrication and microfluidics process are diffusion through a matrix, labour-intensive, high instrument cost, complex design, and requirement of a cleanroom [73,74,75]. These issues need to be resolved for microfluidics to be used on an industrial scale.

## 5. Polystyrene (Organic Hard Template Route)

Hard templating utilises hard particles as a seed. These are then coated via sol–gel chemistry and hollow particles are obtained after the template’s removal. Inorganic, organic, and silica materials can be used to synthesise hollow silica spheres. Inorganic templates have a high interaction with silica, so the coating is high [19,20,21]. However, an inorganic material such as calcium carbonate can give rise to rod-like shapes, so extra care is needed when synthesising a spherical template [21,23]. Inorganic template’s high melting point and their high chemical stability imply that either high temperature or chemicals such as organic solvents, strong acids or bases are required for successful removal, which can be time-consuming and expensive [21,22,23,109]. When designing the process, environmental effects have to be evaluated when toxic chemicals are needed to remove the template. It has been found that the degree of the condensation of silica depends on the reaction parameters, which allow for lowly condensed silica to be used as a template with or without a protective layer, where etching is directed internally [95,110]. The most common etching methods to remove silica are the hydrothermal method [93], sodium borohydride (NaBH_4_) [111], ammonia [93], and sodium hydroxide (NaOH) [110]. Nevertheless, etching processes have sometimes been reported to be incomplete in hard templates [98].

Organic hard templates are preferred to inorganic templates due to their low cost, ease of production, and removal. The template controls the internal diameter while the surface and solution reaction governs the shell growth, allowing for the synthesis of particles with uniform hollow interiors and shell thickness in a batch fashion. Moreover, products with high purity are obtained since the reaction occurs through the sol–gel process. As mentioned in Table 1, although highly promising, the organic template route has low efficiency due to poor surface reaction related to its low surface energy. If the problem of surface energy can be overcome, monodispersed hollow silica with desired porosity, shell thickness, and hollow interior cavity can be produced, which will lower the need for intensive labour and hence the cost. 

PS and polymethyl methacrylate (PMMA) (Figure 8) are commonly used as organic templates because their low surface energy makes synthesising spherical shapes simpler, and their size dispersity can be easily controlled. The removal of organic templates is relatively straightforward as organic materials decompose at lower temperatures than inorganic ones as they are less chemically stable. Initiators and stabilisers are needed to initiate and control the polymerisation process. Even though the low surface energy should make silica deposition difficult, it is shown that the presence of a stabiliser such as poly(vinylpyrrolidone) (PVP) and poly(acrylic acid) (PAA) on the surface can improve the coating [92] as it acts as a nucleation site. A dual template method has been deployed where the organic template controls the internal diameter, and the surfactant controls the shell growth/surface reaction [112]. The surfactant influences the pore structure and acts as a coating aid/linker on the surface of the template. These will be discussed in more detail in the following sections.

The most common procedure for producing hollow silica particles by organic template involves PS spheres in a solution of NH_4_OH, H_2_O, EtOH, TEOS, and a surfactant. TEOS is usually added at the end, either in a one-pot or dropwise fashion. It should be noted that the mechanism of how the template is coated is still under debate: it is either coated directly by hydrolysed TEOS [113] or by silica aggregates formed by condensed TEOS [49,114,115,116]. The last step is the removal of the template, either by calcination or chemical etching, leaving silica spheres with empty cores. The interaction between the template surface and silica particles in the solution must be high for the coating to be uniform and thick. Therefore, understanding the electrostatic interaction, surface energy, nucleation, and reaction kinetics is vital to controlling the shell thickness. An appropriate amount of catalyst, precursors, solvent, water, and a specific addition method are needed for controlled coating (i.e., the silica shell). 

## 6. Polystyrene Coating

Research studies on the effect of coating PS in solutions containing varying amounts of NH_4_OH, H_2_O, EtOH, and TEOS will be reviewed first. If a removal method is not mentioned, the PS was removed by calcination. Most studies investigated the effect of TEOS concentration on the shell thickness. Deng et al. [78] successfully coated and simultaneously removed PS by adjusting the ammonia concentration at a temperature of 50 °C. Low ammonia concentration resulted in either incomplete or no etching of the PS, while no free silica was reported when the ammonia concentration was too high. The complete dissolution of PS was only achieved above the 0.054 g/mL concentration for ammonia and monodispersed hollow silica with no reported free silica particles. (Figure 9) Shell thickness was shown to increase with the increase in TEOS concentration but when it was increased above 0.02 g/mL in 0.03 g/mL ammonia, an increase in free silica particles was also found. At a high concentration of TEOS, a high homogenous reaction is preferred to shell formation.

It is possible to use both aqueous and non-aqueous solutions to produce hollow silica spheres, as shown by Qi et al. [83]. They successfully synthesised hollow silica nanospheres with different PS sizes in a weak base condition. They employed a high PS concentration to promote a heterogeneous reaction. The hollow spheres produced were monodispersed with smooth and ordered shells. An aqueous solution was chosen for PS templates with diameters ranging between 130 and 1500 nm, while a non-aqueous solution was used for 400 nm PS. Hollow silica spheres with diameters of 140, 400, and 1500 nm were produced (Figure 10). This work shows that controlled shell growth can be obtained by controlling the sol–gel reaction in a weak basic condition (ammonia), using an appropriate amount of PS (high), and by utilising both types of solution (ethanol/water). Sharma et al. [91] synthesised hollow silica spheres in a non-aqueous solution and coated them with resorcinol or carbon. This showed that the hollow silica spheres could be further modified.

M. Tan et al. [96] produced coagulated particles, free solid particles, and smooth surface hollow spheres by changing the volume of TEOS and the ratio of ammonia/TEOS. It was established that TEOS between 1.02 and 1.5% vol was suitable for producing hollow silica nanospheres with a good surface. On the other hand, TEOS above 2.08% and below 0.56% vol produced coagulation and no hollow spheres, respectively. The importance of the ammonia/TEOS ratio was shown by the failure of the PS coating when such a ratio was too low.

This was due to the low presence of the catalyst, leading to a slow and ineffective sol–gel reaction. The optimum time for the reaction was suggested to be 10 h and the temperature did not significantly affect the coating of PS. Belostozky et al. [97] synthesised hollow silica spheres via multiple coatings. TEOS was replenished every 18 h to form three layers of silica. Successive coatings increased the shell thickness, but free solid silica particles that formed in the solution were larger than those deposited on PS.

The above works showed that at low TEOS concentration, secondary nucleation was suppressed, as nuclei formed in a later stage and could not grow larger. Therefore, the optimum condition for coating is to have the concentration of TEOS as low as possible. In theory, this leaves more precursors for the PS coating as the competition by free silica should be low. This is a repeating scenario in most of the work reviewed here. A low count or total absence of free silica particles decreases the competition for template coating and improves the coating efficiency. Strategies such as dropwise and multiple one-pot synthesis methods can be used to suppress or lower the effect of secondary nucleation. Both high and low concentrations of ammonia can result in the formation of a shell, but high ammonia is needed if the dissolution of the template is desired. Shell thickness was tuned by either choosing the appropriate TEOS concentration or PS content in a stable system, where the sol–gel reaction is slow enough to be controlled. When the reaction kinetics are slow, the solution’s homogenous reaction is severely reduced or suppressed. The coating was shown to be possible in both aqueous and non-aqueous solutions.

## 7. Surface Modification

As previously discussed, PS has a low surface reactivity due to its low surface energy, but it can be modified either during synthesis or post-synthesis to improve the coating. Understanding the PS surface charge is essential to enhance the electrostatic interaction with ions in solution. A negative surface attracts positively charged ions or particles in solution, while a positively charged surface attracts negatively charged ions or particles. The surface charge density determines the ability to attract ions or particles with the opposite charges.

Surface modification can be achieved via either alternating the chemical composition of the surface or through physical change by modifying the morphology. Surface modification methods of PS range from wet chemical processes to plasma treatment. As an example, the phenyl group of styrene can be modified via organic aromatic chemistry to introduce hydroxyl (–OH), nitro (–NO_2_), amine (–NH_2_) and/or other groups. These functional groups can introduce a negative charge (–OH, –NO_2_) and a positive charge (–NH_2_) on the surface of the PS. Alternatively, it is possible to copolymerise the PS with more reactive monomers such as poly(4-vinylpyridine) [85].

As previously mentioned, during the synthesis of PS, commonly used initiators are AIBN and KPS, and stabilisers are PVP and PAA. Hong et al. [92] used a non-aqueous solution to prepare hollow silica spheres with PS synthesised using AIBN and KPS. AIBN was used with PVP or PAA as stabilisers, while sodium chloride (NaCl) was used as an electrolyte with KPS for surfactant-free polymerisation. No free silica particles were found on the coated shell when PVP stabilised PS was used as a template. In the other combination, when PS prepared with KPS with NaCl and AIBN with PAA were used, free silica was found on the surface of the coated spheres. It was argued that the difference was due to a better reaction between the acidic TEOS and silica particles with basic PVP on the surface of PS, with PVP essentially acting as a nucleation site (Figure 11).

This theory of a nucleation site on PS by the stabiliser was further explored by C. Sandberg et al. [58] They synthesised PS in the nm range with PVP as the stabiliser and KPS as the initiator. They varied the PVP/styrene ratio during synthesis to observe its effect on the coating of PS with silica. The PS diameter was observed to decrease with an increased PVP/styrene ratio. When PS with a PVP/styrene ratio of 0.0075 was used, no silica coating was seen, but this improved when the PVP/styrene ratio was 0.05. It was argued that the partial coverage of PVP at a low PVP/styrene ratio could not shield the negative charge of sulphate from KPS, allowing the repulsion between the negative surface and negative silica. The authors claimed that when enough PVP was present, it could shield the negative charge so the silica’s particles could deposit on the PS. It was suggested that a positive surface charge would provide better results. However, W. Chen et al. [99] produced highly dispersed hollow silica particles using negatively charged PS, where KPS was used as an anionic initiator and acrylic acid (AA) as a co-monomer and a source of negative charge. PS with a different zeta potential was created using different concentrations of AA during PS synthesis, where higher AA gave more negative zeta potential. It was suggested that the negative surface of PS-co-AA attracted ammonia ions in the solution by electrostatic interaction. The electric double layer with positive ammonia at the outer layer attracted negative silica ions (SiO_4_^−^). The charge density of the surface was determined to be the reason for partial or uniform coating and why it was not always possible to synthesise a hollow silica sphere with a negative charge on the surface of PS [98,99]. The surface’s low negative charge density cannot attract enough ammonium ions to create an electric double layer with a positive charge on the outer layer. Due to low surface charge density, the silica deposition will be disproportional because ammonium ions do not cover the whole surface. This produces islands of coating with no continuous shell, a shell too thin, or a non-uniform coating, which, after the removal of the template, fails to generate hollow silica spheres with a uniform shell. Figure 12 shows the effect of the surface charge density on the coating.

The charge effect was further studied by Lu et al. [98]. The work was carried out in propan-2-ol solution. They modified the surface charge by adding –SO_3_ (PS-SO_3_) and –NH_2_ (PS-NH_2_) to make it negatively and positively charged, respectively. They found that a positive surface had a uniform coating, and the negative surface had an irregular coating, which could be due to low negative charge density, as shown by M. Chen et al. [77]. The authors argued that this difference was due to good interaction between the positive surface and negative silica (Figure 13).

The pH of the solution also appeared to have a significant impact where pH <10 produced no coating, while pH between 10 and 11.6 produced a coating with a smooth surface. Above pH 11.6, only a rough surface was seen. The difference in the coating and morphology at different pH was argued to be due to a balance between the homogenous reaction (secondary nucleation) and heterogeneous reaction (surface coating). It was suggested that this pH range could be applied to PS-NH_2_ spheres with different sizes and concentrations as the size of the PS template was found to have a minimal effect on the morphology. It was found that the appropriate TEOS concentration was between 18 mm and 63 mm at pH 11.6.

Good interaction between the positive surface and silica is further supported by Chen et al. [77]. They used 2-(methacryloyl)ethyltrimethylammonium chloride (MTC) as a co-monomer with styrene to produce positively charged 1.2 micron PS-co-MTC spheres. The dissolution of PS was carried out in the same medium by increasing the temperature to 50 °C without further addition or filtration, similar to Deng et al. [78]. No free silica particles were found in the solution. It was observed that at 0.11 g/mL TEOS, 0.045 g/mL ammonia gave a hollow sphere with a smooth surface and PS was completely removed (Figure 14b). Below this amount of ammonia, incomplete etching or no etching was observed (Figure 14a). At low ammonia (0.030 g/mL), a hybrid particle with an unetched template as the core and silica shell was produced. When the ammonia concentration was increased above 0.045 g/mL, dents and deformations were found on the surface of the hollow sphere (Figure 14c). This was argued to be related to a faster dissolution of PS than a sol–gel reaction, which leaves dents on the template where silica will be deposited. At 0.045 g/mL ammonia, hollow silica spheres were reported to have a smooth surface. It was also shown that the wall thickness could be controlled by controlling the TEOS concentration. The wall thickness increased when TEOS was increased from 0.10 to 0.15 g/mL (Figure 14a1,b1). However, when TEOS was increased to 0.18 g/mL, an uneven surface with high side products was seen (Figure 14c1). These results, along with the results by Deng et al. [78], showed that a concentration of ammonia above a certain value could lead to deformed hollow silica spheres as the PS surface could be etched before silica deposition. From the data, it can be concluded that with conditions where tests were carried out, an appropriate ammonia concentration to obtain smooth silica shells would be around 0.035 g/mL.

H. Zou et al. [80] used PVP stabilised PS in non-aqueous solutions to prepare hollow silica spheres, where the PS was removed via dissolution in the same medium. TEOS was added dropwise, and KPS was used as an initiator to produce negatively charged PS. The percentage of PVP on the PS surface increased while the size of PS decreased when the concentration of PVP was raised during the synthesis of PS. It was suggested that PVP on the surface acts as a coupling agent. Low ammonia (0.003 g/mL) resulted in a uniform coating with a smooth surface, while high ammonia (above 0.012 g/mL) produced a rough surface with large amounts of free silica. Rough surfaces were arguably brought about by fast reaction kinetics due to the high ammonia concentration, which also caused high secondary nucleation. Complete precipitation was observed at 0.048 g/mL ammonia, while larger silica particles were seen at higher ammonia concentrations. However, at low ammonia (0.003 g/mL), no hollow structure was formed due to a low dissolution rate, but it was shown that dissolution could be improved by increasing the temperature (70 °C). At this temperature, complete dissolution was achieved, and 0.006 g/mL was the minimum concentration of ammonia needed to produce a hollow silica sphere. It was found that increasing TEOS concentration by dropwise addition made the shell thicker and increased the etching time. However, incomplete etching and composite particles were still seen at higher ammonia concentrations and longer dissolution times. It was concluded that dissolution could be controlled better by adjusting the temperature rather than the ammonia concentration.

It is also possible to introduce silica particles on PS during the synthesis of the PS sphere. M. Takafuji et al. [107] synthesised hollow silica microspheres using a template made from PS and silica nanoparticles modified with 3-methacryloxypropyl trimethoxisilane (M). M molecules were grafted to silica nanoparticles (MSiN) and mixed in an emulsion of styrene, EGDMA (ethylene glycol dimethylacrylate), AIBN, PVA, and water. This creates a PS template with MSiN (hydrophobic) on the surface due to the migration of MSiN to the interface of the suspension droplet from the oil phase (styrene) to the aqueous phase (water) during pre-stirring. This caused MSiN to form a shell while the core was a hybrid between PS and MSiN. MSiN on the surface will interact better with silica particles in the solution than unmodified PS. MSiN-PS particles were then coated with TEOS, followed by calcination to remove the core. Shrinkage of the microspheres was reported after calcination. Figure 15 shows the effect of different concentrations of TEOS on the coating of PS. Similar to H. Zou et al.’s [80] work with different ammonia concentrations, the impact of TEOS concentration on the shell’s growth was reported here. At a high concentration of TEOS (above 0.16 mol/L), irregular, uneven growth was seen, whereas orderly growth was observed at a low TEOS concentration (below 0.16 mol/L). Similarly, after calcination, shell thickness was reported to increase from 0.1 to 1.25 micron with an increase in TEOS concentration. Broken shells were observed up to 0.08 mol/L TEOS. This confirms that the minimum shell thickness is needed to withstand the pressure change due to the decomposition of the PS template. L. Zhang et al. [82] coated PS with AA followed by PAH poly(allylamine hydrochloride (PAH) to give a positive surface charge. When this template was used, hollow spheres with smooth surfaces were produced.

These works show that interaction between silica and the PS surface can be improved by modifying the surface charge with initiators, stabilisers, chemicals, and coating with another polymer with the desired charge. Silica can be introduced on the surface of PS during the synthesis of PS spheres. This shows that the coating can be improved by either electrostatic interaction, introducing nucleation sites and new chemicals with better interaction with silica than PS.

## 8. Effect of Surfactant

A surfactant such as CTAB is widely used to synthesise hollow sphere silica particles (Figure 16) and its concentration was found to have a more significant effect on the pore distribution but not on pore size. Q. Qu et al. [89] used octadecylamine as a surfactant to expand the pore size. G. Zhu et al. [76] synthesised hollow spheres at pH >12 with NaOH using negatively charged PS-SO_3_ and CTAB. It was reported that hollow silica spheres could not be obtained below a pH of 12 with the conditions used. W. Peng et al. [65] investigated the role of CTAB in the coating of PS. PS with 145 nm and 212 nm diameters produced hollow silica spheres with external diameters of 157 and 245 nm, respectively. Shell thickness was again found to increase with the TEOS amount. Porosity test with nitrogen adsorption–desorption isotherms showed that changing the CTAB content did not affect the pore size, but the pore distribution and specific surface area. Results suggest that CTAB plays a vital role in controlling the morphology, specific surface area, and pore volume. X. Wu et al. [61] synthesised hollow silica spheres with a raspberry-like morphology, as observed by SEM (Figure 17). The shell was formed of aggregated small silica particles with a rough morphology and superhydrophobic nature, attributed to the surface roughness. At low TEOS concentration (0.06 mol/L), the shell appeared to be loose and when TEOS was increased (0.12 mol/L), the shell became thicker, and it was mainly formed by small silica spheres packed closely together. As observed in several studies, a high TEOS concentration (0.24 mol/L) can promote homogenous precipitation. The effect of pH at constant TEOS concentration (0.12 mol/L) was investigated and the results, depicted in Figure 18, show that when the pH was below 10, small irregular particles and no raspberry-like morphology were seen. When the pH was between 10 and 12, a protruding surface morphology with interparticle pores was visible but disappeared at pH > 13.2. It was suggested that the smooth shell resulted from a higher degree of homogenous reaction and the generation of small silica particles, which then fused to create a tight coating. It has been argued that CTAB adsorption and pore templating of CTAB is higher at high pH.

The introduction of surfactants will also alter the surface charge, which can have an adverse or positive effect. H. Blas et al. [79] used positively and negatively charged PS with CTAB in both aqueous and non-aqueous solutions to study the effect of the surface charge on the coating. In the study, when an aqueous solution with no ethanol was used to coat PS, monodispersed hollow silica spheres with a similar morphology were successfully synthesised with both surface charges. In the work, KPS was used as the initiator and dihexylsulphosuccinate sodium salt as a surfactant for the negatively charged PS nanospheres, while positively charged PS nanospheres were obtained using 2,2′-azobis(2-amidinopropane) dihydrochloride (AAH) as the initiator and CTAB as the surfactant.

When the coating and calcination were carried out in a non-aqueous solution, both negative and positively charged PS yielded pores with perpendicular orientation. TEOS/PS weight ratio was used to control the shell thickness. Hollow silica spheres with different external diameters were synthesised by changing the PS size and the TEOS/PS weight ratio. No influence of surface charge on the PS coating was observed in a non-aqueous solution, attributed to a slow condensation reaction in a low amount of water. In an aqueous solution, shells were made of an aggregate of small silica spheres for both surface charges, and pores were found to have parallel orientation. The authors suggested that parallel orientation is probably related to fast condensation in an aqueous solution (*no ethanol*). It can also be due to different interactions of CTAB and water due to changes in the solubility of CTAB in the absence of ethanol. This orientation difference could be exploited to produce shells with different porosity. Negative surface charged PS was reported to produce a better surface coating, however, the effect of surface charge was not conclusive due to the smaller size of the positively charged PS used. The size needs to be similar to find a definitive impact of surface charge. Further study of positive surface charge was carried out by Q. Shang et al. [100] where PVP functionalised positive PS was used with AIBA as the initiator. CTAB was used as a surfactant and suggested to be a pore-forming agent during the PS coating. It was proposed that some TEOS molecules were adsorbed on PS and shells were formed from the interaction between the adsorbed TEOS and silica aggregates formed in the solution from the water/TEOS/NH_4_OH/CTAB emulsion. Additionally, the electrostatic attraction was proposed to be another reason for the deposition of silica particles on positively charged PS. When the TEOS concentration was changed from 0.016 to 0.061 g/mL in an aqueous solution with fixed ammonia concentration (≈0.018 g/mL), silica nanoparticles on the surface were found to increase with increases in the TEOS concentration. An intact shell was only achieved after calcination when the concentration of TEOS was above 0.022 g/mL. When the concentration of TEOS was increased to 0.089 g/mL, free silica particles were seen, which indicates a high homogenous reaction. The amount of free silica nanoparticles was found to rise with the increase in TEOS concentration. The average size of surface silica increased with an increase in catalyst concentration, and when NH_4_OH was above 0.028 g/mL, loosely covered PS and broken shells were observed. Results showed that the NH_4_OH/TEOS volume ratio was a significant factor. Q. Shang et al. [100] argued that the adsorption of silica species contributed to the formation of the first layer and was followed by the deposition of silica particles from solution onto the silica layer formed on PS via van der Waals interactions. It has been reported that sufficient condensation cannot occur for a coating at NH_4_OH/TEOS ratio < 1.

C. Ge et al. [81] used PS and methyl acrylic acid (MMA)-functionalized PS (PS-MMA) as a template. PS-MMA and CTAB were found to produce a uniform coating compared to pure PS. This difference in the coating was due to better interaction between CTAB and MMA. Wei et al. [102] used AIBA and AIBN as initiators to prepare PS with a positive and negative surface, respectively, which was confirmed by a different surface potential at different pH by zeta potential analysis. Monodispersed hollow silica spheres with a diameter of ~1.6 microns were successfully synthesised using positive PS with CTAB. The shell was reported to be uniform with a thickness of 150 nm. M. Khoeini et al. [60] used a different approach; initially, solid silica nanoparticles were synthesised with the aid of CTAB and used to coat the negatively charged PS to form a shell. The negatively charged PS was synthesised using KPS as an initiator, and sodium dodecyl sulphate and sodium bicarbonate as the surfactant. Hollow spheres were successfully obtained by coating PS with synthesised silica nanoparticles, taking advantage of the interaction of the CTAB micelle, surface charge, and negative silica particles. Differential thermal analysis (DTA)/thermogravimetric analysis (TG) were performed on CTAB and was found to decompose between 200–300 °C, which suggests that composite PS with only a silica shell could be synthesised with CTAB completely removed at above these temperatures.

Li et al. [90] used tetramethylammonium hydroxide (TMAOH) as a catalyst and methyltrimethoxysilane and 3-mercaptopropyltrimethoxy-silane as the precursors. Sodium lignosulphonate (SLS) and PVP were used as surfactants to successfully synthesise small hollow SiO_2_ spheres, which were used to coat larger PS templates. TMAOH was used in the syntheses of small hollow silica particles as well as in the formation of shells made up of small hollow silica particles on the PS template, similar to M. Khoeini et al. [60] Tetrahydrofuran (THF) was used to selectively etch the PS template from the silica spheres with a raspberry-like morphology and high porosity. PVP and SLS were found to act as a linker between nanoparticles and PS surfaces. Depending on the concentration, they were found to promote/inhibit the formation of free silica microspheres. Y. Guo et al. [103] synthesised a hollow silica sphere where silica nanoparticles modified with 3-(trimethoxysilyl) propylmethacrylate was encapsulated. Initially, modified silica nanoparticles were embedded in PS-co-AA and then used as a template. The coating was carried out with the aid of CTAB in an aqueous solution. Compared to unmodified PS, which yielded a non-uniform shell, a smooth uniform shell was achieved with PS-co-AA as a template. This difference was reported due to poor interaction between CTAB and the unmodified PS surface. High interaction between silica particles and the PS-co-AA surface was attributed to electrostatic attraction between cationic CTAB and the negative charge from the carboxylic group of AA. As reported in several works [62,63,64], CTAB was proposed to act as a linker between the template surface and silica particles in the solution. When the CTAB concentration was 0.0013 g/mL at pH 9, a high amount of free silica particles and a uniform coating were seen. A smooth and uniform shell was obtained at a 0.0008 g/mL concentration of CTAB, and no free silica was observed in the solution. Wang et al. [85] approached the synthesis of hollow silica spheres differently. A catalyst was introduced on the surface of the template by creating PS-co-P4VP (poly(4-vinylpyridine)) composite spheres where some of the P4VP exposed on the surface acted as a catalyst. Since no additional catalyst was used, the reaction only occurred on the surface of the template with the help of CTAB (Figure 19). The main issue with this setup is the shielding effect of the shell once it is formed. This halts the reaction by impeding the interaction between the precursor and catalyst site. As the mass ratio of template/TEOS was increased, the shell thickness was observed to decrease, and monodisperse hollow silica particles were successfully synthesised without any additional catalyst. H. Fan et al. [87] explored the effect of positive and negative charged PS with CTAB in a non-aqueous solution and observed that coating on negative PS gave an irregular shape. In contrast, a positive PS yielded a uniform shell with a rough surface. Shell thickness was found to be proportional to the TEOS/PS ratio, while surface area and pore volume were inversely proportional. Additionally, shrinkage was observed after calcination.

These works clearly show that the coating can be controlled by adjusting the surfactant concentration, and it has been suggested that they act as linker and pore-forming agents. It is unclear whether the seen effect is simply due to adsorption on the surface or via the formation of micelles in solution, which provides nucleation sites for silica, which then deposits onto the surface of PS. Even though the mechanism is not well-understood, the effect of surfactants on the coating can be exploited. The surfactant must be optimised as its concentration can also adversely affect the shell thickness and morphology. High or low surfactant concentration can lead to undesired results such as no coating, non-uniform coating, or high amounts of side products. Effectivity of the surfactant on the coating can be further improved by introducing specific functionalities on the PS template by using specific co-monomers or initiators during its synthesis. This would result in a stronger electrostatic/chemical interaction with the surfactant or surfactant/TEOS micelles.

## 9. Supercritical Fluid

A supercritical fluid is a substance where the liquid and gas phases cannot be distinguished above its critical pressure and temperature. The high kinetic energy above critical temperature allows molecules in the supercritical fluid to overcome the intermolecular forces to change into the gas phase. However, the high pressure causes molecules to condense. Due to low viscosity and high diffusion rate, supercritical fluids can diffuse through solids and transport materials in or out of the solids. Supercritical CO_2_ can be used to extract caffeine from black tea, which has its use in the coffee and tea industries [117]. Z. Chen et al. [108] used supercritical CO_2_ to prepare hollow silica spheres using cross-linked PS-MA-EGDMA (MA = methacrylic acid) microspheres. These spheres were treated with supercritical CO_2_ in a stainless autoclave containing TEOS, and after depressurisation of the system, the PS-TEOS template was coated via a sol–gel reaction. It was proposed that PS swells in a homogenous phase of supercritical CO_2_ and TEOS, where TEOS diffuses through PS and gets trapped inside, forming a composite material. Therefore, the sol–gel reaction is directed toward the surface due to trapped TEOS diffusing to the surface. In contrast to pure PS, PS-MA-EGDMA was found to resist deformation when immersed in supercritical CO_2_. The shell of silica formed inside and outside of PS is illustrated in Figure 20, and the shell thickness increased with an increase in the TEOS concentration. It was concluded that parameters such as time and TEOS concentration during supercritical CO_2_ treatment affected the coating greatly. 

Similarly, G. Sun et al. [88] successfully used two types of crosslinked PS (*PS-EGDMA and PS-DVB* (*divinylbenzene*)) that were treated with supercritical CO_2_ to produce hollow silica. Crosslinked PS-EGDMA and PS-DVB with an average diameter of 220 and 400 nm were treated with TEOS in a stainless-steel autoclave at different pressures with supercritical CO_2_. After depressurisation, the treated PS was coated with a sol–gel reaction with ammonium hydroxide as a catalyst. Coated PS-EGDMA (240 nm external diameter) and PS-DVB (420 nm external diameter) produced hollow silica spheres with a 20 nm and 24 nm shell thickness, respectively. Similar to Z. Chen et al. [108], trapped TEOS was suggested to contribute to the shell formation. The diffusion rate of trapped TEOS to the PS surface and reaction kinetics in solution was proposed to determine the shell growth. Even after forming a layer of silica on the surface of the PS, TEOS trapped inside continued to contribute to the growth of the inner shell until complete consumption. Coating without supercritical CO_2_ treatment resulted in a strawberry-like morphology with higher surface roughness. An increased number of incompletely coated PS was reported, resulting in a decrease in hollow silica spheres after calcination. Shell thickness was controlled by adjusting the TEOS/PS ratio during the coating of the template. The pressure and time of the supercritical CO_2_ treatment were vital for forming a uniform shell, and PS-EGDMA, treated with a pressure between 10 and 12 MPa with 2 mL of TEOS, produced a 25 and 40 nm shell thickness, respectively. Treatment time between 1 and 4 h at 10 MPa was tested with PS-EGDMA, and the results varied from cracked shells to thicker shells. This indicated that TEOS needed time to diffuse into the template, but beyond the 2-h treatment, the outcome was not clear. A shell transition from 20 nm to broken was observed when TEOS was reduced from 4 mL to 1 mL. This shows that a threshold concentration of TEOS is required during supercritical CO_2_ to achieve a stable shell. The role of diffused TEOS in solution was seen when the TEOS concentration was increased from 0.03 to 0.08 g/mL. The shell thickness increased from 40 nm to 60 nm, but solid spheres were seen at 0.08 g/mL. Even though the shell thickness increased with the TEOS concentration, the homogenous reaction also seemed to increase.

Using supercritical CO_2_, hollow silica spheres were successfully synthesised. It has been shown that TEOS can penetrate both PS and cross-linked PS, where the latter is a better substrate for supercritical treatment due to its ability to resist deformation. The level of TEOS penetration can be controlled by changing the pressure and exposure time and trapped TEOS migrated to the surface and reacted inside the PS with few “sticking” out of the surface during the sol–gel reaction. This created a shell inside and outside the template. The interaction between silica and silica was much higher than with PS, resulting in thicker coatings.

## 10. Other Factors

The main aim of this review was to summarise recent progress on the coating of PS. The focus was on the concentration of catalyst, precursor, surfactant, surface charge, and the modified surface of PS. In addition to these, other factors have been reported to influence coatings of PS in other works, which are briefly described below.

### 10.1. Effect of Type of Solution

Hollow silica spheres have been achieved in both aqueous and non-aqueous solutions [118]. Non-aqueous solutions are preferred due to the slower reaction kinetic, allowing better control over the sol–gel reaction. However, Kato et al. [84] conducted systematic research on the stability of hollow silica particles with PS at different CTAB concentrations and EtOH volume fractions at a 0.0056 g/mL ammonia concentration and 0.0029 g/mL TEOS concentration. It was found that depending on the CTAB concentration, the stability of the hollow sphere was dependent on the volume fraction of EtOH. High and low concentrations of CTAB produced either a higher amount of side products or broken shells during the coating process. The relation between CTAB concentration and volume fraction of EtOH for the stability of hollow silica was studied in an aqueous solution. When CTAB was 5.5 mM, it was found that low volume fractions of ethanol (between 0.19 and 0.25) produced collapsed hollow silica spheres, while volume fractions of 0.31 and 0.37 were able to make monodispersed hollow silica spheres. Figure 21 shows areas where monodispersed hollow silica spheres failed to produce a monodispersed hollow silica (including broken shells and fused particles). In Figure 21, area (i) represents no hollow silica sphere, area (ii) represents broken shells, area (iii) represents fused particles, and area (iv) represents a monodispersed hollow silica sphere and can be described with the formation of non-uniform, thin, and thick shells. Shell thickness was found to increase with the increased volume fraction of ethanol, which was attributed to lower hydrolysis of TEOS in a high ethanol volume fraction. This results in low nucleation and slower growth of solid silica, thus lower by-products. This means less competition for the reaction between PS and silica so that the shell can grow thicker.

L. Yin et al. [62] studied the effect of CTAB with negative PS in an aqueous solution. With ammonia (NH_3_) fixed at 0.0178 g/mL and CTAB concentration at 0.0162 g/mL, changing the ETOH:H_2_O ratio led to varying products such as hollow spheres with a large hole and PS sandwiched with silica (*uneven coating, high aggregation*) (Figure 22). At a 0.13 EtOH:water solution, a high number of small particles were seen surrounding the template. This is due to the fast hydrolysis and condensation of TEOS as the homogenous reaction is favoured. Silica particles were smaller in a ratio EtOH:H_2_O of 0.13 than in 0.51. On the other hand, at an EtOH:H_2_O ratio of 3.8, silica particles grew between two adjacent PS to produce PS–SiO_2_–PS as a result of slow reaction kinetics. The effect of the concentration of NH_3_ in an aqueous solution was also investigated and increasing the concentration of NH_3_ from 0.0179 to 0.515 g/mL at 0.0162 g/mL CTAB and 0.51 EtOH: H_2_O resulted in a transition from a small hole on the hollow spheres to non-uniform large hollow spheres. An increase in non-uniformity on the surface of PS with an increase in ammonia was proposed to be due to fast hydrolysis and condensation, causing solid silica particles to grow larger before deposition. It was suggested that CTAB has a dual purpose where it acted as a template for the mesopore and a linker to silica. Electrostatic interaction between negatively charged silica nanoparticles on PS/CTAB improved the deposition rate, as the surfactants acted as linkers due to their cationic nature. Results suggest that an adequate covering of PS with CTAB is required to achieve a full coating. These findings were supported when 0.0041 g/mL CTAB in 0.51 EtOH:H_2_O and 0.0179 g/mL ammonia produced particles with irregular hollow cavities and broken shells, suggesting a thin coating and irregular shells. The main cause for this was suggested to be an insufficient CTAB coverage of the PS surface. Increasing CTAB to 0.0081 g/mL produced a hollow sphere with uniform morphology and a smaller cavity. No effect on the pore size was seen when varying CTAB concentration, and results showed that the interaction of the PS surface and CTAB increased with an increase in the amount of CTAB.

### 10.2. Effect of Temperature in Calcination and Aging

Calcination is preferred to chemical etching because of the chemical hazards commonly used, and etching can be incomplete if not carefully controlled. M. N. Gorsd et al. [104] found that when toluene was used along with calcination to remove PS, it had an adverse effect, and broken shells were observed. Zhang et al. [105] demonstrated that the Stöber method produced a silica network where the concentration of silicon atoms with three (Q3) to four (Q4) –O–Si– units (Figure 5, Q (040)) was high. It was found that a rearrangement within the network occurred when calcination was conducted at a high temperature, leading to a fully condensed (Q4) silica network. Q4 content was found to increase compared to Q3 as the temperature increased. P. Ruckdeschel et al. [106] reported a decrease in microporosity and surface area when the calcination temperature was increased from 500 °C to 950 °C. This could be due to sintering. C. Ge et al. [81] showed the effect of temperature on aging on porosity where PS-MAA, CTAB, TEOS, and ammonium solutions were used. It was found that aging at room temperature gave microporous hollow silica while aging at 150 °C resulted in mesoporous hollow silica spheres. Therefore, the calcination and aging temperature must be carefully selected to control the porosity.

### 10.3. Effect of pH

Ammonium hydroxide is used widely as a catalyst for the generation of silica via the sol–gel process due to its low cost and ease of handling, but it is highly volatile. Therefore, alternative catalysts have been tested during the modified Stöber method. G. Zhu et al. [76] used NaOH to synthesise hollow silica spheres successfully, while J. Wang et al. [119] replaced the ammonia with lysine, n-butylamine, and n-propylamine to synthesise solid silica nanoparticles. M. Meier et al. [35] used ethanolamine as a catalyst and successfully synthesised solid monodispersed silica nanoparticles. Ethanolamine was found to have higher activity and stability than ammonia with temperature. Yamashita et al. [120] used a mixture of ammonia and sodium hydroxide, where the PAA emulsion was used (soft template). Results suggest that sodium hydroxide primarily acted as a catalyst for silica coating on the template, while ammonia affected the PAA aggregation. A mixture of catalysts was recommended to catalyse the silica reaction in the solution and onto the surface of the template.

As seen from the sections above, many variables contribute to coating efficiency; therefore, a large set of experiments is required to fully understand the effect of different catalysts on the coating of PS.

## 11. Applications

The application of hollow silica spheres vary and depend on their size as well as the porosity and thickness of the shell. They share some properties with solid silica spheres such as high surface area, the same chemical properties, and the ability to be functionalised [121,122,123], but with a hollow interior and permeable shell, allowing them to have controlled absorption and release of chemicals. Moreover, hollow silica spheres can be used in lightweight fillers, low-dielectric materials, drug carriers, catalyst support, etc. [124,125,126,127]. Some hollow nano/microsphere applications and future perspectives are discussed below.

The hollow cavity has been shown to have lower thermal conductivity than solid silica spheres, silica aerogels, wool, glass fibres, and PS foams [58,61,128,129]. This is because conduction is low as solid is only present in the shell, and air, which is a poor thermal conductor, is present in the core. Filling the hollow silica with low thermal conductivity gases can further improve the insulation. Therefore, the insulation properties of hollow silica spheres depend on the size and shell of the sphere as a thicker shell and smaller cavity will have lower insulation compared to a larger cavity and thinner shell. Phenomena such as solid conduction and collision between air molecules and the shell wall contribute to the insulation difference. A higher collision between the shell and the air molecules results in higher energy conduction. The thicker shell provides high solid conduction, and the smaller cavity results in higher collision between air molecules and the cavity wall, also known as the Knudsen effect. Optimising the shell and hollow cavity dimensions is crucial to achieving the desired insulation effect. The surface morphology and purity of hollow silica spheres also play a role in the insulation where rough surface, low solid silica particles on the shell, and unbroken/uniform shells provide better insulation. Hollow silica nanospheres can be used to make structures with low thermal conductivity [12,129] and additives to polymers such as polyethersulfone where thermal conductivity was drastically reduced [10].

Hollow silica nanospheres have also been proposed as an alternative to delivering drugs to a specific organ. The biocompatibility of silica compared to other nanoparticles such as carbon nanotubes makes it a good candidate since it can be removed from the body and has a longer circulation lifetime [130]. A hollow cavity and a porous shell allow them to be loaded with different chemicals on the surface and in the core, and a controlled diffusion can be achieved in different scenarios. Controlled diffusion has been demonstrated by surface modification or shell thickness [131,132]. It has been shown that they can deliver proteins, vaccines, and contrasting agents for medical imaging [14,133,134]. A buoyant hollow silica microsphere with an antibody on its surface has been used to separate pathogens from stool, which was a quicker and cheaper method than the current techniques for cell separation when an isolation test of Cryptosporidium parasite from stool was carried out [135].

Hollow silica spheres have been shown to improve catalytic activity. Silver nanoparticles were encapsulated in a hollow silica nanosphere, which led to a longer lifetime of the catalyst. It could be reused after heat treatment and higher adsorption was seen when tested with hydrogen sulphide and tert-butylmercaptan [136]. Hollow silica nanospheres with encapsulated silver were produced using polystyrene grafted with AA. This resulted in high catalytic activity for reducing 4-nitrophenol to 4-aminophenol by NaBH_4_ [137]. D. Li et al. [67] tested the detoxification capability of hollow silica microspheres filled with ethyl butyrate as oil to remove iodine from an aqueous solution. Since there is a low iodine affinity with silica, it was expected to interact highly with the oil core. The UV absorption intensity of the solution decreased up to 95% after 30 s of adding hollow silica microspheres, indicating the quick removal and high efficiency of the detoxification of the filled hollow silica microspheres compared to liquid–liquid extraction. Moreover, the reusability of microspheres was shown, which could reduce the cost of the entire process, and it was concluded that a specific waste could be extracted or removed from a solution by using an appropriate filling in the core.

As a nanoreactor, hollow silica nanospheres were used to successfully synthesise hollow tin oxide (SnO_2_) nanospheres, which were shown to have uniform size, structure, and high lithium capacity retention [138]. E. Cevik et al. [139] reported high charge–discharge cycle reversible capacity and stable cycle performance when multi-shelled hollow silica microspheres were used as an anode in lithium batteries compared to solid silica particles. Sulphonated hollow silica particles filled with sulfuric acid were used as an additive to a PVA matrix, increasing the energy storage efficiency. This resulted in a supercapacitor with a high energy density of 20.40 W·h kg^−1^ at a power density of 545 W·kg^−1^. 

The inner cavity of a hollow sphere will determine its loading capacity. Compared to conventional nanospheres, hollow silica microspheres allow for higher loading capacity and tuneable porosity, and surface modification improves their functionality. This will be useful in energy storage, waste removal, detoxification, and microreactor, but further improvement in synthesis and functionalisation is needed to bring the hollow silica microspheres to commercial viability.

## 12. Summary

Various research studies covering the synthesis of hollow silica spheres using PS as a template were reviewed and the reasons for the popularity of hard templates, especially PS, were explained. Key parameters affecting the Stöber reaction, which affects the coating, were briefly described to explain the importance of controlling the reaction kinetics. Since both homogenous and heterogeneous reactions can occur in the solution, it is clear how vital it is to promote the heterogeneous reaction and stop or suppress the homogenous reaction to improve the coating efficiency. Several methods were presented to increase the coating by choosing optimum parameters and introducing new chemicals that enhance the interaction between silica and the PS surface.

Most of the work reviewed here shows a general trend and the key findings are summarised below:Controlling the TEOS concentration is the best way to control the reaction kinetics and have high efficiency in the coating. TEOS concentration should be as low as possible to suppress the secondary nucleation by starving the system from the precursor, so the amount of free solid particles would be low. Most of the TEOS will be consumed by the PS surface as there will be less competition. However, if too low, a uniform coating cannot be achieved as there is insufficient TEOS to cover the surface, while a high TEOS concentration results in an increased number of free solid silica particles.The catalyst has been pointed out as a primary factor in controlling the sol–gel reaction and affecting the coating of PS. The concentration and strength of the catalyst play a vital role in controlling pH as well as the reaction kinetics of the sol–gel and has been shown to affect the structure by influencing the degree of hydrolysis during the induction period. Hollow silica spheres have been produced in low and high pH, depending on the other chosen parameters.Initiators and stabilisers used during the synthesis of PS have been shown to significantly affect the coating by providing electrostatic attraction and nucleation sites, respectively. Functional groups can be introduced during template synthesis or later via coating or chemical treatment to produce the desired surface charge. Both negatively and positively charged PS has been used to make hollow silica spheres. It has been suggested that positively charged PS would be better for coating as the coating’s negatively charged surface is greatly affected by the charge density.Surfactants have been shown to improve the coating significantly and it has been suggested that they act as a linker between the template surface and silica. It also has been recommended to act as a pore structuring agent. Surfactant concentration affects the pore distribution but not the pore size. This can be used to control the porosity. The morphology, shell thickness, and pore distribution can be tuned by controlling the surfactant concentration. The surfactant concentration was shown to affect the stability of hollow silica, and this effect could also be tuned by varying the ethanol:water ratio in the solution.Chemicals such as polymers, silica particles, and other reactive chemicals can be introduced onto the surface of PS during or after the synthesis of spheres to improve the coating. After the synthesis of PS spheres, reactive chemicals can be introduced by coating. Subjecting it to supercritical CO_2_ can make the PS template vulnerable to penetration by TEOS. The overall aim of these methods was to have more reactive chemicals on the surface, which will have better interaction with silica in the solution than PS.

Other parameters such as solution (*ethanol: water ratio*), catalyst type, and calcination temperature were briefly reviewed. Due to the interactive effects of parameters in a sol-gel reaction, the results from various works are in poor agreement with each other due to the chosen reaction conditions. For these reasons, extrapolating the data could result in a failure of the coating process as minor changes might significantly alter the stability of the system. Additionally, when larger PS in the micron range requires coating, the surface energy needs to be considered as nanomaterials have a much higher surface energy than micromaterials. Future work aiming at modifying one of the components such as the catalyst should therefore consider that it may require a completely different set of parameters such as the water/ethanol ratio, surfactant concentration, etc. to produce a successful coating. Therefore, a large set of experiments would be required to achieve the optimum conditions.

## Figures and Tables

**Figure 1 materials-15-08578-f001:**
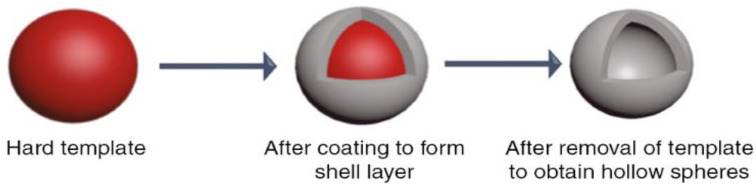
Hard templating route for hollow silica [16].

**Figure 4 materials-15-08578-f004:**
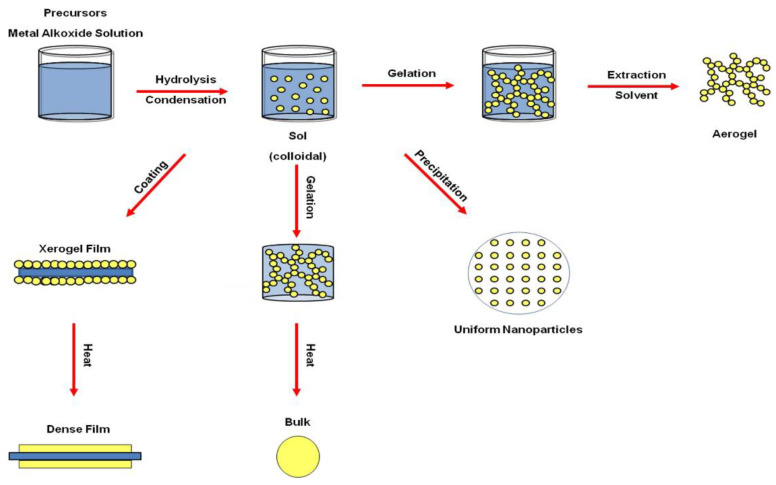
Different routes of the sol–gel reaction [34].

**Figure 6 materials-15-08578-f006:**
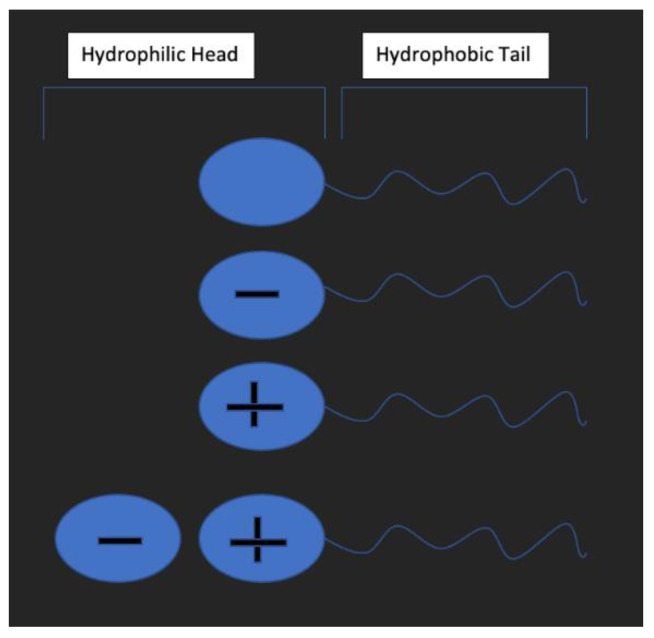
Types of surfactants according to charge.

**Figure 7 materials-15-08578-f007:**
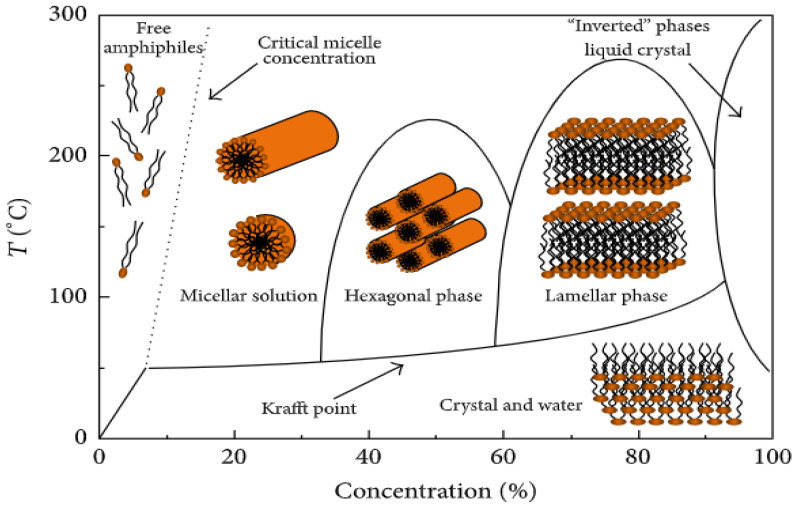
Graph of the effect of the surfactant concentration on the surface tension and their phase evolution [66].

**Figure 8 materials-15-08578-f008:**
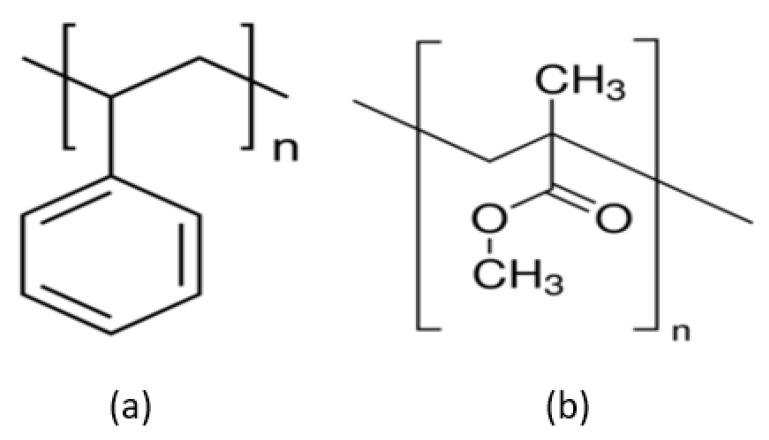
Monomer of (**a**) polystyrene and (**b**) polymethyl methacrylate.

**Figure 9 materials-15-08578-f009:**
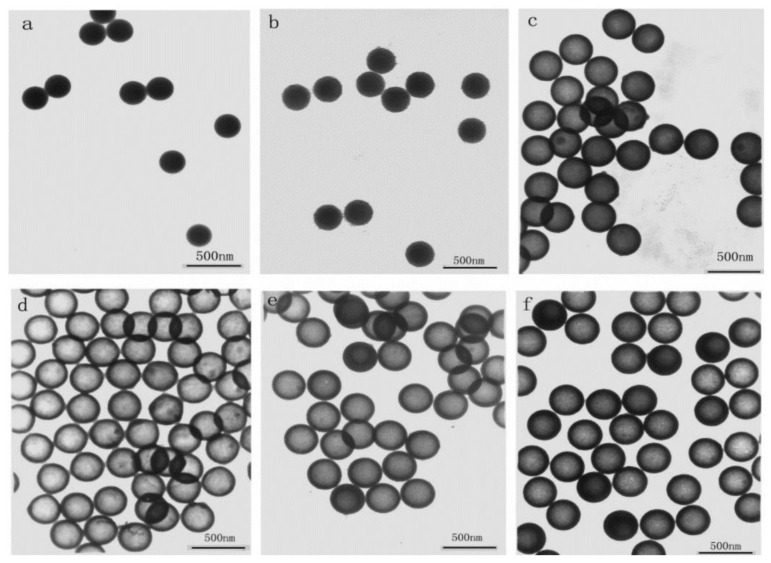
TEM images of (**a**) PS spheres and coated spheres obtained at different amounts of ammonia, (**b**) 0.5 mL, (**c**) 2.0 mL, (**d**) 3.0 mL, (**e**) 4.0 mL, and (**f**) 5.0 mL [78].

**Figure 10 materials-15-08578-f010:**
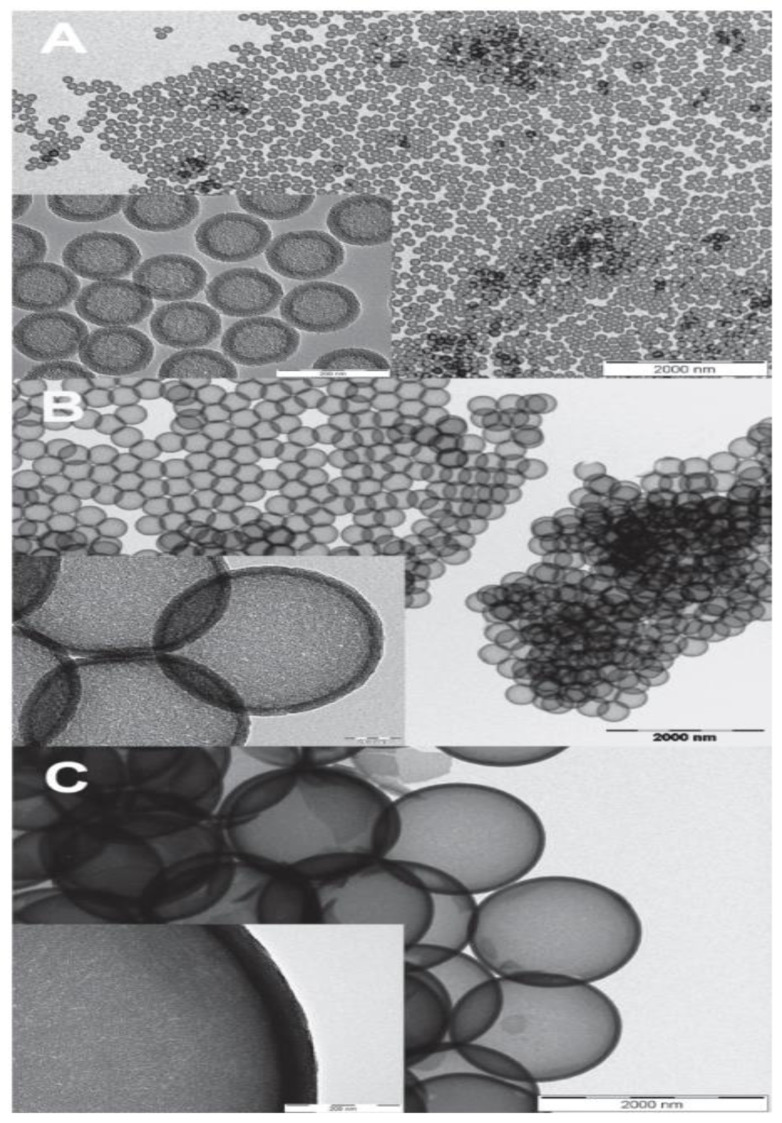
TEM images of a hollow silica sphere with different silica particles. The contrast between the core and shell shows the absence/presence of a hollow cavity. The size of PS was (**A**) 140 nm, inset scale bar = 200nm (**B**) 400 nm, inset scale bar = 100nm and (**C**) 1500 nm, inset scale bar = 200 [83].

**Figure 11 materials-15-08578-f011:**
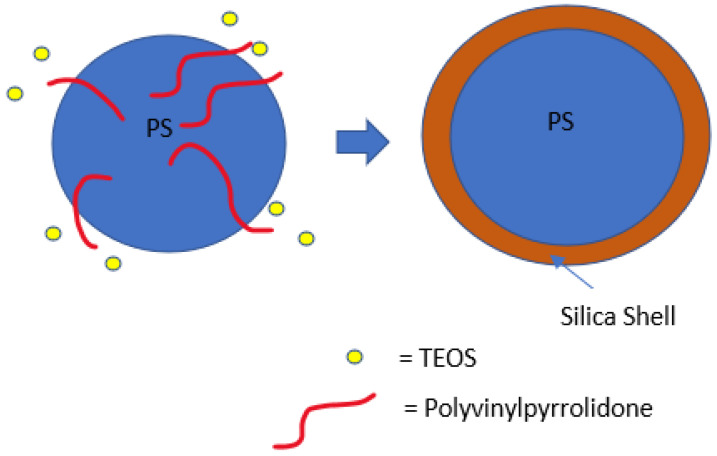
Synthetic route from PS to shell formation on PS due to nucleation of TEOS at PVP. (Adapted from Hong et al.) [92].

**Figure 12 materials-15-08578-f012:**
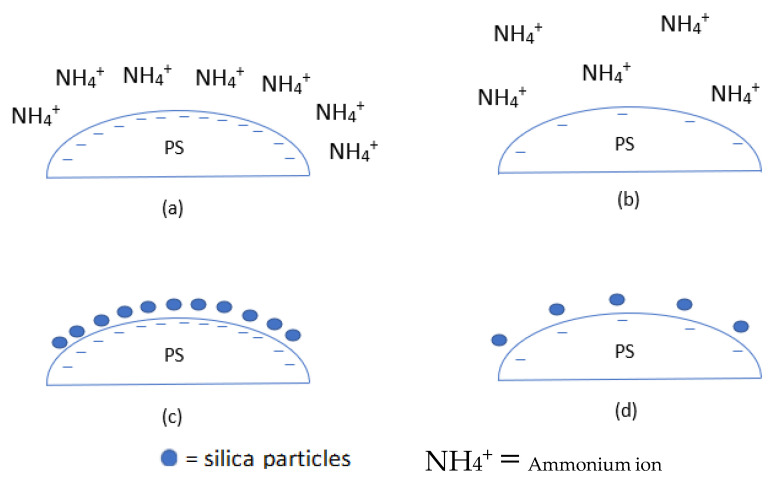
Schematic of the effect of the negative surface charge density on coating. (**a**,**c**) High surface charge density and (**b**,**d**) low surface charge density.

**Figure 13 materials-15-08578-f013:**
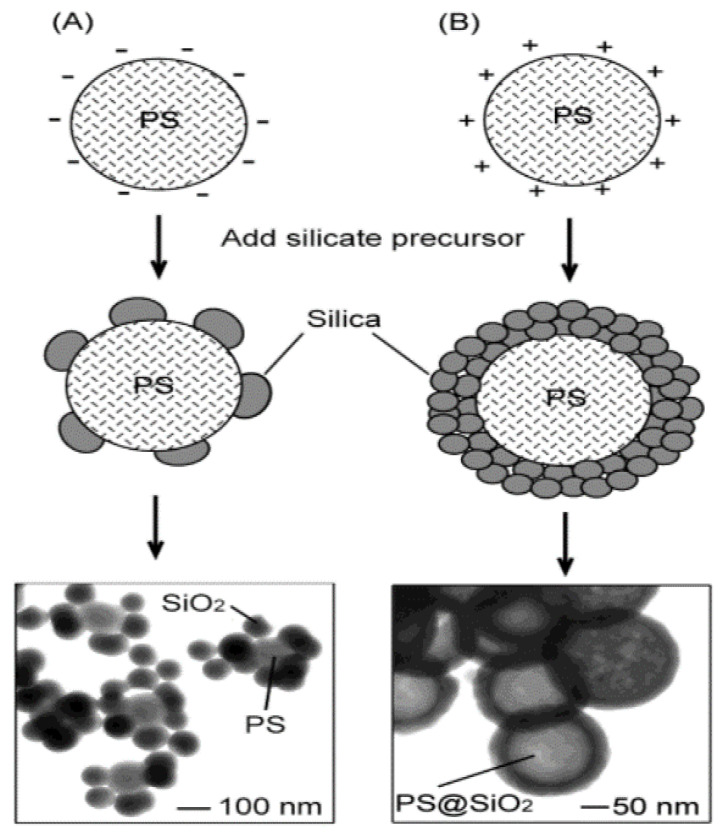
Schematics illustrating the effect of PS beads terminated in (**A**) the –SO_3_ and (**B**) –NH_2_ groups. The TEM images of the PS coatings obtained are also shown [98].

**Figure 14 materials-15-08578-f014:**
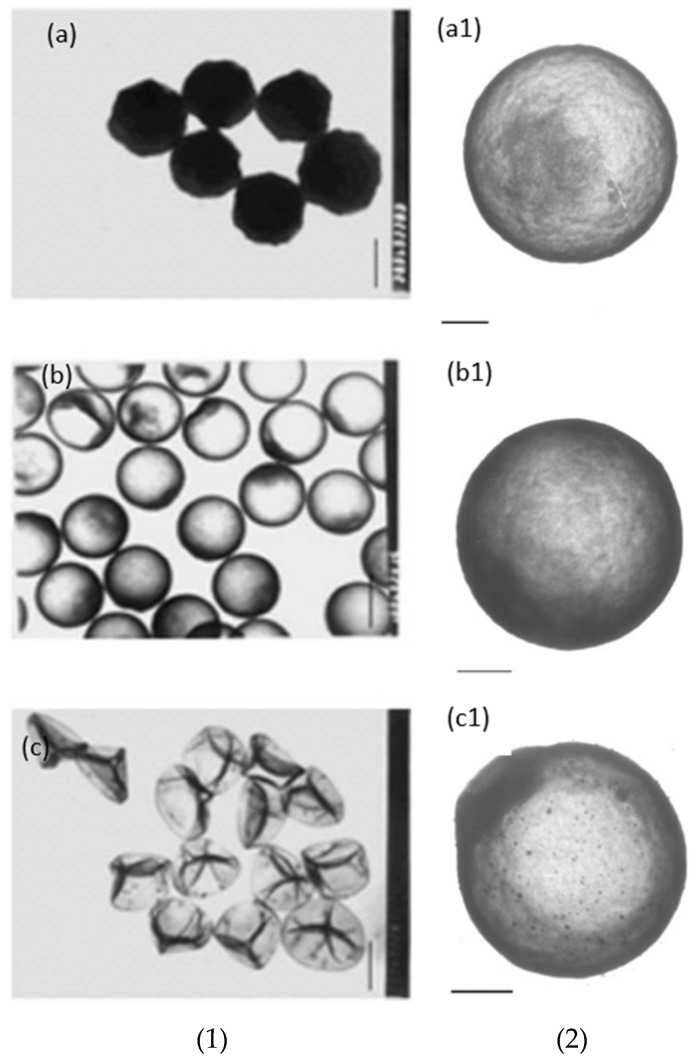
(1) TEM images of silica spheres prepared at different amounts of ammonia of (**a**) less than 0.030 g/mL, (**b**) 0.045 g/mL, and (**c**) more than 0.045 g/ml; (2) TEM images of hollow silica prepared at 0.045 g/mL ammonia with different amounts of TEOS: (**a1**) 0.10 g/mL, (**b1**) 0.15 g/mL, and (**c1**) 0.18 g/mL. The scale bar is 1 μm (modified original image) [77].

**Figure 15 materials-15-08578-f015:**
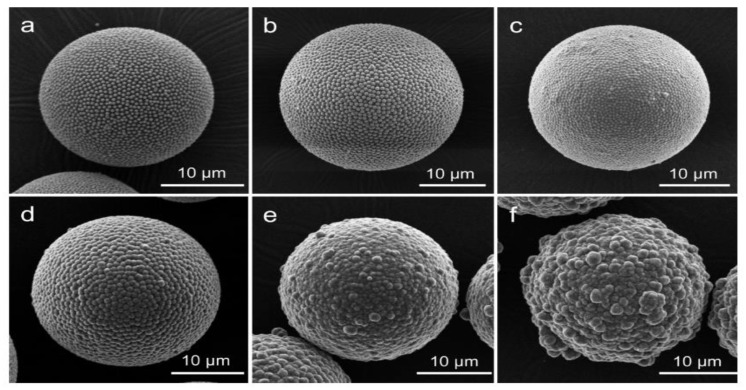
SEM images of the coated MSiNPS at different TEOS volume. (**a**) No TEOS, only MSiNPS, (**b**) 0.04 mol/L, (**c**) 0.08 mol/L, (**d**) 0.16 mol/L, (**e**) 0.32 mol/L, (**f**) 0.48 mol/L [107].

**Figure 16 materials-15-08578-f016:**
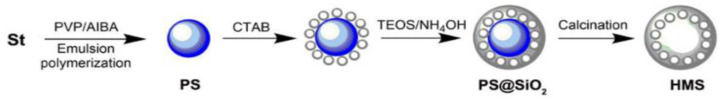
Schematic diagram of the synthesis of hollow mesoporous silica spheres (HMSs) using the surfactant and PS [65].

**Figure 17 materials-15-08578-f017:**
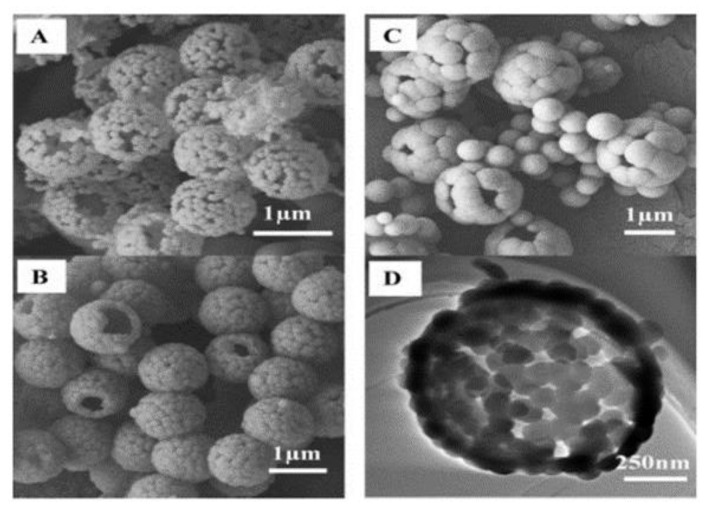
SEM images of hollow silica at pH 10 at different TEOS concentrations: (**A**) 0.06 M, (**B**) 0.12 M, (**C**) 0.24 M. (**D**) TEM of hollow silica spheres achieved at a TEOS concentration = 0.12 M [61].

**Figure 18 materials-15-08578-f018:**
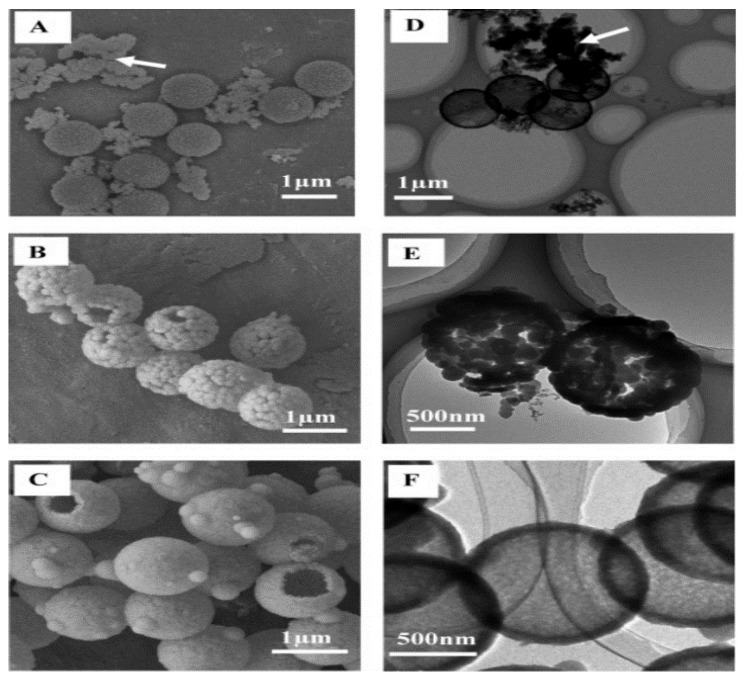
SEM (**A**–**C**) and TEM (**D**–**F**) images of the hollow silica when the TEOS concentration = 0.12 M and different pH. (**A**,**D**) pH 9, (**B**,**E**) pH 12, and (**C**,**F**) pH 13 [61].

**Figure 19 materials-15-08578-f019:**
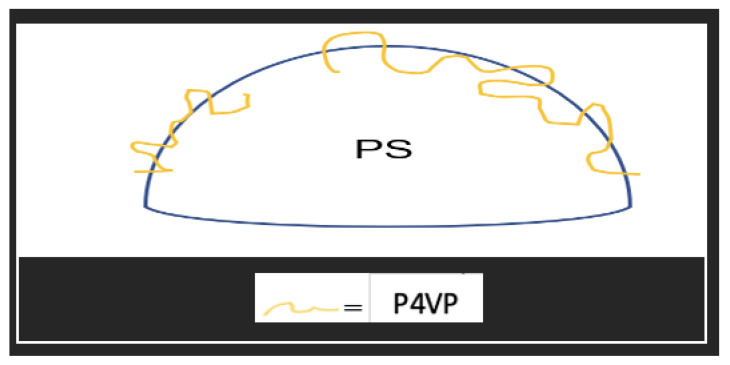
Schematic of PS-co-P4VP with P4VP (poly(4-vinylpyridine)) on the surface.

**Figure 20 materials-15-08578-f020:**
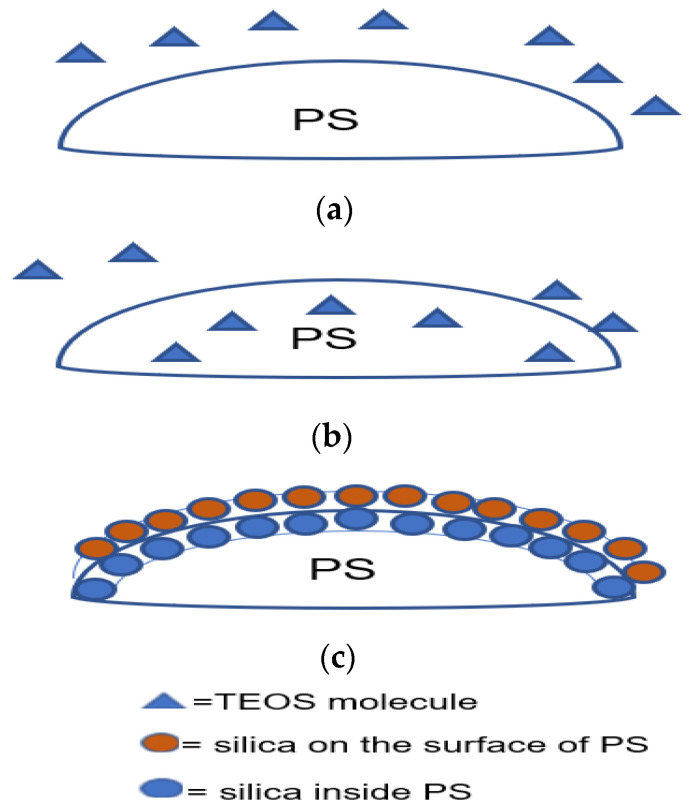
Schematic of the formation of silica on PS with the aid of supercritical CO_2_. (**a**) PS and TEOS before supercritical CO_2_ treatment, (**b**) penetration of TEOS inside PS after immersion in supercritical CO_2_, and (**c**) formation of the shell of silica inside and outside PS after the sol–gel reaction.

**Figure 21 materials-15-08578-f021:**
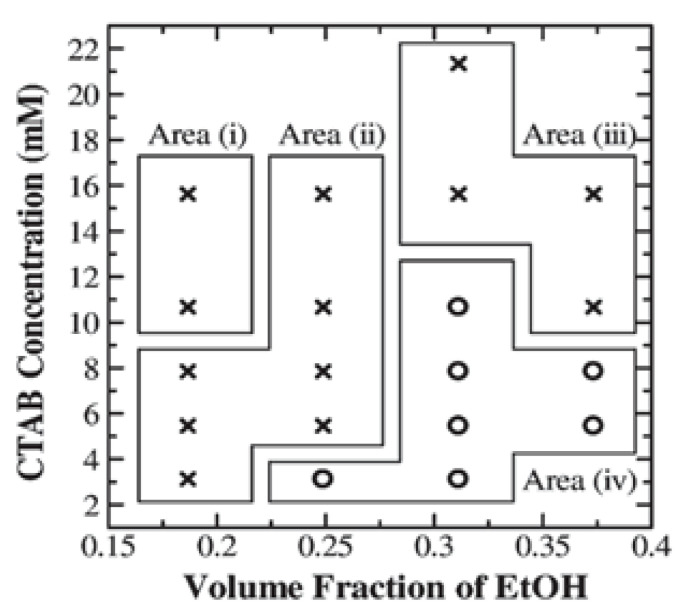
Effect of CTAB concentration on the stability of hollow silica spheres in an aqueous solution. “o” represents uniform monodispersed hollow silica spheres while “×” represents failed hollow silica sphere, which includes fused or aggregated hollow silica spheres [84].

**Figure 22 materials-15-08578-f022:**
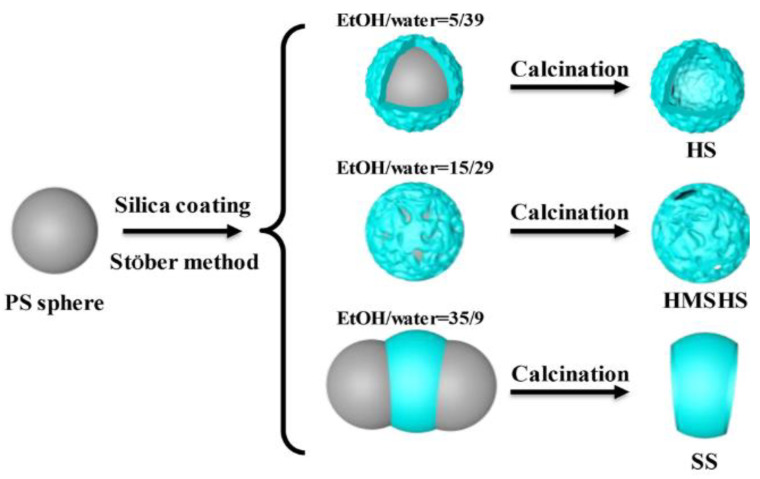
Schematic diagram of the effect of EtOH/water on the morphology. HS = hollow silica, HMSHS = large through-holes on the shell, and SS = sandwich-like silica [62].

**Table 1 materials-15-08578-t001:** Comparison of the various methods of synthesising hollow silica nanoparticles [15].

No.	Strategy	Advantages	Disadvantages
1	Polymer micelles/emulsions Surfactant examples: CTAB	Good for synthesis of small HSP’s Well-established synthesis process	Less control of particle size Low yield
2	Inorganic template Example: Carbon	HSP’s size control	Incomplete dissolution of inorganic core Time-consuming High cost Low yield
3	Organic template Example: Polystyrene	HSP’s size control Well-established process	Low yield High cost Solvent wastage
4	Solid silica particle etching	HSP’s size control Well-known chemistries	Less control of hollow cavity size Time-consuming
5	Spray drying method	Potential for scale-up	Less HSP size control Needs preformed silica nanoparticles
6	Spray pyrolysis	Potential for scale-up	Polydisperse HSP’s
7	Bacteria/virus templates	A wide range of available template shapes	Costly Less scalable Less HSP size control

CTAB = Cetyltrimethylammonium bromide, HSP = Hollow silica particle.

**Table 2 materials-15-08578-t002:** List of hollow silica spheres with the average diameter, shell thickness, and pore size.

No.	Template	Precursor	Catalyst	Average Particle Size (nm)	Shell Thickness (nm)	Pore Size (nm)	Reference
1	PS	TEOS, TMOS	NaOH	NA	NA	1.2	[76]
2	PS/CTAB	TEOS	NH_4_OH	NA	62, 24	2.7	[59]
3	Positive PS	TEOS	NH_4_OH	NA	50, 100	NA	[77]
4	Positive PS	TEOS	NH_4_OH	NA	20, 30, 45	24.3, 37.4	[78]
5	PS/CTAB	TEOS	NH_4_OH	NA	NA	2.7	[61]
6	PS/CTAB	TEOS	NH_4_OH	90,200	5 40	2.3, 1.12	[79]
7	PS	TEOS	NH_4_OH	NA	15, 40	NA	[80]
8	PS-co-PMMA, CTAB	TEOS	NH_4_OH	350	30	1.8, 2.71	[81]
9	PS(PAH/PVP)	TEOS	NH_4_OH	NA	15, 70	NA	[82]
10	PS/CTAB	TEOS	NH_4_OH	NA	12, 24, 51	3, 3.6	[83]
11	PS/CTAB	TEOS	NH_4_OH	425, 767	57, 142	NA	[84]
12	PS-co-P4VP/CTAB	TEOS	P4VP	375	12, 26, 39	2.7, 5.4	[85]
13	PS, CTAB	TEOS	NH_4_OH	45, 90	NA	2.27, 2.19	[86]
14	PS/CTAB	TEOS	NH_4_OH	NA	15–95	NA	[87]
15	PS-EGDMA, PS-DVB/trapped TEOS	TEOS	NH_4_OH	240, 420	15–60	NA	[88]
16	CTAB/ODA	TEOS	NH_4_OH	300	NA	1.9, 2.0	[89]
17	PS	TEOS	NH_4_OH	2500	70	2.8,4.1	[90]
18	PS/CTAB	MTM/MPTMS	TMAOH	2400	25	macropores	[60]
19	PS/CTAB	TEOS	NH_4_OH	157, 245	14.1, 19	2.2, 2.3	[65]
20	PS	TEOS	NH_4_OH	152, 228	14, 19	2.3	[91]
21	PS/CTAB	TEOS	NH_4_OH	NA	NA	7.1, 22.3	[62]
22	PS-PVP/PAA	TEOS	NH_4_OH	1000–2320	50–70	NA	[92]
23	SiO_2_	TEOS	NH_4_OH	45–450	30	2.5, 3.2, 3.4	[93]
24	SiO_2_	TEOS	NH_4_OH	73–100, 330, 333, 418, 2600	5, 14–18, 41, 66, 88	3.6, 5.5	[94]
25	SiO_2_	TEOS	NH_4_OH	430	NA	NA	[95]
26	PS	TEOS	NH_4_OH	350–450	NA	NA	[96]
27	PS	TEOS	NH_4_OH	2500	70	NA	[97]
28	PS	TEOS	NH_4_OH	NA	50–150	NA	[98]
29	PS- AA	TEOS	NH_4_OH	NA	15, 20, 25	NA	[99]
30	PS/CTAB	TEOS	NH_4_OH	320	5, 32.8, 58.4	5.04	[100]
31	PSS-MAA/CTAB	Na_2_SiO_3_	NH_4_OH	362, 421, 750	18, 51	1.78, 5.57	[101]
32	PS/CTAB	TEOS	NH_4_OH	1500	150	8.36	[102]
33	PS-AA	TEOS	NH_4_OH	540, 1700	10–20, 40	NA	[103]
34	PS	TEOS	NH_4_OH	NA	100	NA	[104]
35	PS-PAH-PVP	TEOS	NH_4_OH	1520–1810	1590–1830	NA	[105]
36	PS	TEOS	NH_4_OH	310–316	43–44	NA	[106]
37	PS + MSiN	TEOS	NH_4_OH	20,000	100–1250	NA	[107]
38	PS/trapped TEOS	TEOS	NH_4_OH	325	60	7.03	[108]

PS = polystyrene, TEOS = tetraethyl orthosilicate, TMOS = tetramethyl orthosilicate, NaOH = sodium hydroxide, CTAB = hexadecyltrimethylammonium bromide, PVP = polyvinylpyrrolidone, PAA = polyacrylic acid, AA = acryclic acid, PMMA = poly(methyl methacrylate), PAH = poly(allylamine hydrochloride), P4VP = poly(4-vinylpyridine), EGDMA = ethylene glycol dimethylacrylate, DVB = divinylbenzene, ODA = octadecylamine, MTM = methytrimethoxysilane, MPTMS = 3-mercaptopropyltrimethoxysilane, TMAOH = tetramethylammonium hydroxide, MSiN = 3-methacryloxypropyl trimethoxisilane modified silica nanoparticle, NA = not available.

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
