# Peer review of "Hollow Silica Nano and Micro Spheres with Polystyrene Templating: A Mini-Review"

_materials, 2022, doi:10.3390/ma15238578_

Round 1

Reviewer 1 Report

In this manuscript, the authors simiply reviewed the hollow silica nano & micro spheres with polystyrene templating. Overall, the related interpretations look reasonable. Nevertheless, a revision would further improve the quality of this article. I recommend to publish this manuscript with major revision.

1.     Authors mentioned the first synthesis method to create silica nanospheres was the Stöber method (8) in Line 29. Please explain what the meaning of the (8) is?.

2.     The paper is a bit hard to read, one reason is the density of citations and jump from fact to fact, but other is a bit of not very clear English.

Author Response

1) it was an editing error, it has been fixed and made superscript

2) some alteration was made to the manuscript to make it easier to read

Reviewer 2 Report

In this manuscript, Gurung et al. comprehensively summarize the synthesis of hollow silica nano-/micro- spheres using hard templating methods. Template materials, surfactants, solutions and various dominant effects are well discussed in the review. Then, the applications of the silica spheres are briefly described to highlight the importance of the silica nano-/micro- spheres. I would like to recommend the acceptance of the review to publish in Materials after minor revision.

1. The overall structure and logic of the review need to be further improved. For example, only one subtitle “5.1.2 Polystyrene” is inside “5.1 Organic template” catalogue, and only “5.1 Organic template” is under “5. Hard templating”. Under “6. Surface modification”, it is inappropriate to juxtapose the 3 subtitles “6.1 Surface charge and chemical modification”, “6.2. Effect of surfactant”, and “6.3 Supercritical fluid”. The author may redefine the subtitles with better logic.

2. During the introduction section, the authors might briefly discuss the applications of silica hollow spheres to emphasize the importance of related research.

3. Before the "1.2. Comparison of synthesis strategy" section, the authors may provide a brief discussion about "why different synthesis strategy is required" and "why to highlight hard template strategies" to highlight the importance of synthesis strategies.

4. The authors may provide more discussion about the role of supercritical fluid like how it works during the preparation of silica hollow spheres. The authors may consider merging “Supercritical fluid” section into “Effect of solution”.

5. "Effect of PH" might be a more accurate title to summarize the paragraphs in 7.3.

Author Response

Hi,

As requested, we have altered the manuscript as follows:

1) In sections 5 and 6, subsections were either merged or they were assigned their own section.

2) a sentence was added where hollow silica spheres have shown to have their application

3) Information was added on why a hard template is the main focus of this review

4) a brief description of supercritical fluid and its application was added

5) section title was changed as suggested

Round 2

Reviewer 1 Report

It's OK to accept.